# Soil Humus, Iron, Sulphate and Magnesium Content Affect Nectar Traits of Wild Garlic (*Allium ursinum* L.)

**DOI:** 10.3390/plants10030597

**Published:** 2021-03-22

**Authors:** Alexandra Bodó, Ágnes Farkas, Dávid U. Nagy, Kinga Rudolf, Richárd Hoffmann, Marianna Kocsis, Tamás Morschhauser

**Affiliations:** 1Institute of Biology, Faculty of Sciences, University of Pécs, 7624 Pécs, Hungary; alexandrabodo88@gmail.com (A.B.); davenagy9@gmail.com (D.U.N.); morsi@gamma.ttk.pte.hu (T.M.); 2Department of Pharmacognosy, Faculty of Pharmacy, University of Pécs, 7624 Pécs, Hungary; agnes.farkas@aok.pte.hu; 3Institute of Plant Production Science, Campus of Szent István, University of MATE, 7400 Kaposvár, Hungary; rukinga@freemail.hu (K.R.); erwiniar@gmail.com (R.H.)

**Keywords:** *Allium ursinum*, medicinal plant, nectar sugar concentration, nectar volume, soil parameters

## Abstract

Recent studies revealed that from various ecological factors influencing nectar yield and quality of a plant, soil properties can be as important as microclimatic features. To date, few studies have investigated the relationship of soil characters to nectar traits of bee pollinated plants growing in natural associations. Our study intended to reveal which soil properties had the most powerful impact on nectar variables of wild garlic (*Allium ursinum* L.). Specimens were collected from fourteen habitats in two different years, and were potted in their original soil under the same climatic conditions. Nectar volumes and sugar concentrations were measured and soil samples were analysed for fourteen parameters. Statistical analyses revealed that the number of nectar producing *Allium* flowers, as well as the nectar volume and sugar content of nectar in individual flowers were influenced by both year and habitat. The humus, iron and sulphate content of soil showed negative correlation with the number of flowers producing nectar; total nectar volumes were negatively correlated with humus and iron content, but positively affected by magnesium content of the soil. Our results suggest that in addition to the effect of microclimatic factors, certain soil properties can have significant impact on nectar traits.

## 1. Introduction

Environmental factors can have a profound effect on plant performance, characterized by traits like biomass production, chemical composition etc. Similarly, nectar volume and composition is thought to be influenced by several ecological parameters, such as microclimatic conditions (air temperature, relative air humidity, evapotranspiration, amount of solar radiation, wind speed) and soil features like moisture level, temperature and aeration e.g., [1,2,3,4,5,6,7,8,9,10,11]. A recent study revealed that interactions between mānuka (*Leptospermum scoparium* J.R. et G. Forst) cultivars and soils affected plant growth and flowering, and in turn overall nectar yield was influenced by cultivars-soil interaction [12]. Nectar volume, concentration and sugar composition can also vary between flower developmental stages, floral sexes [13], flower position in raceme [14], flower life-span, morphological and phenological characteristics of individual plants [7,15,16,17,18]; and are influenced also by extrinsic factors such as pollinator behavior (frequency and abundance of pollinator visits, the number of flowers visited per plant, the duration of the visit) [13], presence of nectar robbers, or nectar contamination by yeasts [18].

Wild garlic or ramson (*Allium ursinum* L.) is a bulbous perennial herbaceous monocot, widely distributed in mesic, deciduous woodlands of Europe [19], as well as in certain regions of Asia [19,20] and Africa [21]. The species has been used traditionally both for food and medical purposes, such as lowering blood pressure and cholesterol level [22,23]. Besides, a unifloral honey can be obtained from the nectar produced by the flowers. The chemical composition of the plant seems to be largely influenced by the ecological conditions of the habitat, as well as by the biological traits of the plant [23,24,25]. Since the development of reproductive organs and secretion of nectar have a significant metabolic cost for the plant [26], the production of nectar is likely to be highly sensitive to various ecological factors. Our previous study [27] demonstrated that nectar volumes and sugar concentrations of the plant vary in different natural associations, which could be the result of different microclimatic and soil factors. In order to exclude the effect of microclimatic conditions during the blooming period and to focus on the influence of soil properties, in this study *Allium* plants were kept in different types of soil, but under the same microclimatic conditions. We addressed the following questions: (i) which nectar traits of *A. ursinum* are affected by sampling site/soil properties; (ii) whether the effect of soil features is the same in each year; (iii) which soil properties have the greatest influence on the number of nectar producing flowers, and the volume and sugar content of nectar in individual flowers. Our hypotheses were the following: (i) between-site differences in soil characteristics affect the number of wild garlic flowers producing nectar, the amount of nectar produced and the sugar content; (ii) the effect of soil factors can be modified by annually changing climatic factors.

## 2. Results

### 2.1. Effect of the Habitat on Ratio of Nectar-Producing Flowers, Nectar Volume and Concentration in Allium ursinum

The ratio of *A. ursinum* flowers producing nectar ranged from 0.0 to 82.8%, and from 3.9 to 55.7% in the first and second year of the study, respectively (Table 1). We found that both sampling years and sites, and even their interaction influenced this parameter significantly (Figure 1). In addition, the ratio of nectar producing flowers was influenced stronger by the site (df = 7; χ^2^ = 69.043; *p* < 0.001 ***) than by the year (df = 1; χ^2^ = 15.477; *p* < 0.001 ***). Differences between sites could be attributed mainly to the fact that at some sites a large proportion of flowers did not produce any nectar at all. When analysing the relationship of nectar volume in individual flowers to the years of measurement, a highly significant relationship was found (df = 1; χ^2^ = 83.641; *p* < 0.001 ***), while a minor yet significant difference was evident with respect to the sites (df = 7; χ^2^ = 14.233; *p* < 0.05 *) and the interaction (df = 7; χ^2^ = 18.325; *p* < 0.05 *) (Figure 2). There was a single case (1st year Site 7 and Site 14) when the difference was perceptible (Tukey test: t-value = 3.750; *p* < 0.05 *) between two sites in the same year, and in another case the relationship (between 1st year Site 9 and Site 14) was near-significant (Tukey test: t-value = 3.303; *p* = 0.053). However, significant differences in the total nectar volume of individual plants could not be detected between either the sites or the years. The results showed a highly significant (df = 1; χ^2^ = 26.501; *p* < 0.001 ***) difference between the years with respect to average nectar volume, which ranged from 0.1 to 5.5 μL flower^−^^1^. *A. ursinum* flowers produced an average of 0.74 ± 0.71 μL nectar flower^−^^1^. In the first year, the highest average nectar volumes were observed in plants from hornbeam-oak forest, humid ravine forest, and decomposed oak-ash-elm gallery forest. If comparing the nectar volume data of sites examined in both years, two outstanding records were measured from the first year (Site 10 and 14), although these differed significantly only from the second-year data compared to other sites, while in the first year they did not. In some sampling sites (Site 7, 8, 11, 12), significant differences in nectar volume in different years were due to the fact that mostly in the first year, virtually no nectar was produced by individuals selected by random sampling. There was a highly significant difference between the two years regarding the sugar content of nectar of individual flowers (df = 1; χ^2^ = 57.867; *p* < 0.001 ***) (Figure 3), and a lower but statistically relevant difference was also found among the sites (df = 7; χ^2^ = 23.776; *p* < 0.01 **). Moreover, sugar concentration of nectar differed with respect to the various sites in the same year, but only in the second year. In the case of Site 8 and 11 in the first year, solute concentration could not be measured due to very low (or no) nectar volumes, so these were not included in the analysis. In addition, the interaction ‘Year x Site’ was highly significant (df = 7; χ^2^ = 41.428; *p* < 0.001 ***). Mean sugar concentration of nectar produced by *A. ursinum* plants was 30.5 ± 6.04% and 34.09 ± 3.92%, ranging from 10.0 to 45.0% and from 25.0 to 47.0% in individual flowers, in the first and second year, respectively. Mean sugar concentrations of nectar in individuals were significantly different in the two years for most of the sites (df = 1; χ^2^ = 9.946; *p* < 0.01 **), except for Site 10 and Site 12. No significant difference was evident with respect to the sites (df = 7; χ^2^ = 6.106; *p* = 0.527), but the ‘Year x Site’ interaction was highly significant (df = 7; χ^2^ = 24.176; *p* < 0.001 ***). 

### 2.2. Effect of the Soil Parameters on Number Of Nectar Producing Flowers, Nectar Volume and Concentration in Allium ursinum

Among the soil parameters, humus, iron and sulphate (Appendix A) showed negative correlation with the number of producing *A. ursinum* flowers. The relationship was minor yet significant regarding humus content (df = 1; χ^2^ = 4.283; *p* < 0.05 *, r^2^ = 0.303), while highly significant with respect to iron (df = 1; χ^2^ = 12.376; *p* < 0.001 ***; r^2^ = 0.371) and sulphate (df = 1; χ^2^ = 11.662; *p* < 0.001 ***; r^2^ = 0.402) (Figure 4A–C). The number of flowers producing nectar differed nearly significantly with respect to pH (KCl) (df = 1; χ^2^ = 3.318; *p* = 0.069) and phosphate (df = 1; χ^2^ = 3.741; *p* = 0.089) (Figure 5A,B). A statistically relevant negative correlation was also found between ratio of nectar producing flowers and sulphate (df = 1; χ^2^ = 6.428; *p* < 0.05 *; r^2^ = 0.352), as well as between ratio of nectar producing flowers and humus (df = 1; χ^2^ = 7.063; *p* < 0.01 **). A near-significant relationship was evident with regard to iron (df = 1; χ^2^ = 4.360; *p* = 0.064). If soil parameters were related to corresponding total nectar volumes (of individuals), significant relationship was found regarding humus (df = 1; χ^2^ = 6.332; *p* < 0.01 **; r^2^ = 0.362) and iron (df = 1; χ^2^ = 6.752; *p* < 0.01 **; r^2^ = 0.298), and lower but relevant correlation was detected with magnesium (df = 1; χ^2^ = 6.162; *p* < 0.05 *; r^2^ = 0.247) and sulphate (df = 1; χ^2^ = 5.152; *p* < 0.05 *; r^2^ = 0.310). All observed correlations were negative, except for magnesium, which showed positive correlation (Figure 6A–C). Mean nectar volume showed minor yet significant correlation merely with pH (KCl) (df = 1; χ^2^ = 5.271; *p* < 0.05 *; r^2^ = 0.185), whereas mean nectar sugar concentration was related to neither of the soil parameters. Nearly significant differences were found in the case of mean sugar concentration of nectar with respect to pH (H_2_O) (df = 1; χ^2^ = 2.749; *p* = 0.097), phosphate (df = 1; χ^2^ = 2.900; *p* = 0.1), and sulphate (df = 1; χ^2^ = 2.823; *p* = 0.1). Low correlation found between soil properties and nectar traits may be the result of limited number of replications. In case of the other soil factors analysed (PA; soluble salts; Ca, Cu, K, N, Na, Zn content), no statistically significant results were obtained (Appendix A). We also searched for possible correlations among soil elements that showed significant relationship with nectar traits. We concluded that highly positive correlation was found among sulphate, humus and iron, while magnesium correlated significantly only with iron. 

## 3. Discussion

The knowledge of potential nectar yield and composition as the function of microclimatic and edaphic factors is particularly important in the case of nectar sources that provide the basis of large scale honey production, such as black locust (*Robinia pseudoacacia* L.) or oilseed rape (*Brassica napus* L.) in the temperate climate zone. Another important subset of plants worth investigating are nectar plants that provide less frequent kinds of honeys, which might have unique properties, due to the presence of specific active compounds in their nectar and/or honey. The latter group includes *A. ursinum*, a medicinal plant occurring in several natural plant associations in temperate regions.

Comparing nectar production of *A. ursinum* in this study to earlier measurements conducted in Hungary [27], we measured higher maximum values of nectar volume (5.5 μL flower^−1^) in the current study, compared to the previous one (3.8 μL flower^−1^). At the same time, the nectar was less concentrated in the present study, averaging 27.00 ± 12.30% with extreme values 1–47% in 2013–2015, compared to an average of 36.57 ± 3.62%, with extreme values 25–55% in 2012. This finding might be explained by the fact that our previous study [27] was conducted at the original habitats of the plants, where the role of transpiration could be more prominent, resulting in lower amounts of more concentrated nectar in *Allium* flowers.

The dependence of *A. ursinum* on nutrients is well known from literature, both in the case of habitats with abundant water supply [25], and habitats with no excess water [30,31,32]. These papers focus on the relationship between soil nutrients and the morphology or biomass of the whole plant (particularly the leaf and bulb), however, they do not discuss nectar properties.

From the factors influencing the nutrient supply of soil, suitable pH (H_2_O) for wild garlic is thought to be between 4.2 to 8.1, the optimal range being pH 6–7 [33]. This can explain the positive correlation found between pH and the number of nectar producing flowers, which was statistically significant in case of pH (KCl), the latter being more reliable in case of forest soils.

Several studies reported that addition of low amounts of nitrogen to the soil may enhance flower formation, which will result in higher number of flowers per plant [34,35,36,37,38]. However, in *A. ursinum* the number of inflorescences did not increase after supplying additional nitrogen alone, either in cultivated [32] or natural stands [39]. Our study found that nitrogen did not affect the nectar producing capacity or nectar sugar concentration of wild garlic flowers, either, similarly to the case of *Linum lewisii* or *Potentilla pulcherrima* [35,40]. However, in the related species *A. porrum* NPK-fertilization had a positive effect on nectar production [41].

In our study soil phosphate content was found to have a nearly significant positive effect on the number of nectar producing flowers and nectar sugar concentrations. This can be related to the increase in the number of wild garlic inflorescences on the effect of phosphate [30], which is also valid if supplemented together with nitrogen [39], particularly in case of phosphate-limited habitats [42]. The increase in the number of inflorescences was also explained with the prolonged vegetation period, due to climate changes [39], which results in a longer interval when nutrients are available and higher allocation levels to the bulb.

Our study revealed that total nectar volume of individual *A. ursinum* plants did not differ significantly between study sites or years; while nectar sugar concentrations of individual flowers differed significantly between sites and years. The quantity of secreted nectar and the sugar concentration of nectar were found to be year-dependent also in *Campanula patula* [13] and *Hyacinthus orientalis* [14], but in the case of *Oenothera* species no difference was measured within a species regarding total quantity of nectar production by a flower between years [18]. Contrary to our results, the study conducted by [43] on *Croton macrostachyus* L. revealed that there is a significant difference in the mean nectar volume among study sites, but mean nectar concentration was not significantly different among study sites. These results suggest that nectar traits can be affected in a species-specific manner both by study site, and yearly changes of climatic and soil factors.

In an experimental setup similar to ours, when mānuka cultivars were grown on various soils under constant and controlled environmental conditions, Nickless et al. [12] demonstrated that flowering phenology and nectar yield were significantly different on various soils. The mānuka cultivars showed significantly greater growth in response to increased nutrients and some cultivars increased floral density, also suggesting higher nectar yield. The study revealed that sulphate, manganese and chloride content influenced the flowering period; while iron, manganese and calcium concentration affected the number of flowers. Similarly, in our study we pointed out the effect of iron, together with humus and sulphate, on the number of nectar producing flowers, and in turn on nectar production, particularly nectar volumes. An increase of the above three soil factors brought about the decline of the number of nectar producing flowers and total nectar volumes. Although no relevant correlation was found with respect to producing intensity of flowers and nectar sugar concentrations in habitats with higher humus, iron and sulphate content, in these populations fewer flowers were able to produce nectar, and total nectar volumes were lower, which on the whole had a negative effect on nectar production. On the other hand, soil magnesium content positively affected total nectar volume, while it did not affect mean nectar volume.

In our study higher levels of some nutrients had negative effect on nectar production, which can be due to the fact that soils of natural plant associations where wild garlic flourishes are originally rich in nutrients. In the scenario of cultivation, soils are usually lower in nutrients, and increasing their level artificially will result in higher nectar yield [12]. However, further studies would be required to reveal if increasing nutrient levels further would have negative or positive effect on plant growth and in turn nectar yield.

The findings of our study can be useful for beekeepers and producers of ramson honey. Besides species related nectar traits, edaphic and microclimatic characters should be taken into consideration when selecting plant associations with *A. ursinum* where beehives should be placed. Our results suggest that the flowers of *A. ursinum* can provide maximal nectar yields when the soil conditions of the habitat are optimal for nectar production, which does not necessarily mean that conditions are optimal for the plant on the whole. In addition, the yearly climatic differences may have a substantial impact on the nectar producing capacity of plants. Changing microclimatic conditions can alter utilization of nutrients and in turn the nectar producing capacity of plants even in the same habitat. Wild garlic habitats with greater diversity, such as ravine forests, alder gallery forests and transitory oak-hornbeam forests, can be expected to provide nectar for honeybees more evenly and steadily compared to monodominant wild garlic associations in oak-hornbeam and beech forests.

## 4. Materials and Methods

### 4.1. Study Sites, Selection of Plants

*Allium* samples were collected from fourteen natural sites from the Transdanubian area in Hungary in April of 2013 and 2015 (Figure 7, Table 1). Sampling sites included mesic deciduous forests (oak-hornbeam, beech and ravine forests) and alluvial forests (hardwood gallery forest with oak-ash-elm), according to Kevey and Borhidi et al. [28,29]. These plant associations provide different ecological conditions for *A. ursinum*, but the species is dominant in each association. The results of the first year revealed which habitats (sites) were similar regarding edaphic conditions. In the second study year (2015) we excluded sites with highly similar conditions, and reduced the number of study sites to eight, so that each type of plant association was represented by a single site, except the hornbeam oak associations, which slightly differed in their ecological conditions, such as water supply, soil parameters and exposure.

At each study site, three and four plants in the same phenophase (flower bud stage, prior to anthesis) were selected randomly in 2013 and 2015, respectively. Each plant, together with their original soil, was potted separately. Afterwards plants were kept under the same climatic conditions in a growth room. No artificial illumination was applied, the plants received natural light through a window, without direct irradiation. Plants were watered with rainwater to keep their soil moist. Pots were rotated on a daily basis to ensure the same conditions. Our aim with this arrangement was to keep the plants from all sampling sites under the same light, temperature and humidity conditions, in order to focus on differences caused by various soil types from the original habitats. Before nectar analyses, plants were allowed to adjust to the new conditions for seven days.

### 4.2. Soil Analysis

In order to analyse the effect of soil parameters on nectar production, complete soil analysis protocol was performed on samples taken directly from around the roots of examined plants. Soil samples were collected from altogether 13 natural populations of *A. ursinum* in 2013 (Table 1). From each site three soil samples were taken, from 25–30 cm depth and were analysed in parallel. Before analysis, each sample was grinded, air-dried and cleared of plant parts, debris and macroscopic living organisms. Soil samples were sieved with a 2 mm mesh size sieve. Soil analysis was carried out according to the Hungarian Standards at the Accredited Soil Laboratory (104/2015/LAB/NÉBIH) of Kaposvár University, Hungary. The investigated soil parameters included upper limit of plasticity according to Arany (PA), pH (KCl), pH (H_2_O), calcium carbonate (CaCO_3_), organic matter (humus), water soluble salts (salinity), nitrogen (N) content [nitrite (NO_2_) + nitrate (NO_3_)], iron (Fe), potassium oxide (K_2_O), magnesium (Mg), manganese (Mn), phosphorus (P), zinc (Zn), sulphate (SO_4_), and copper (Cu). The N, Mg and sulphur (S) content of the soil was determined out of the fraction obtained with 1 M KCl treatment. The N content (measuring NO_2_+NO_3_) was determined spectrophotometrically at 540 nm with a flow injection analyser (FOSS FIAstar 5000), according to the standards ISO 14255; ISO 14255:1998. The Mg and S (SO_4_) content was determined with inductively coupled plasma (ICP) optical spectrometer (ULTIMA 2 JY ICP OES, Paris, France) according to the Hungarian standard MSZ 20135:1999. The P (PO_4_), K (K_2_O), Na and Ca content of the soil was determined out of the fraction obtained with ammonium lactate treatment with an ULTIMA 2 JY ICP OES spectrometer. The Fe, Zn, Cu and Mn content of the soil was determined out of the fraction obtained with EDTA treatment, measured with ICP according to the Hungarian standard MSZ 20135:1999. For detection of various elements, soil samples were destructed by nitric acid (HNO_3_) and hydrogen peroxide (H_2_O_2_) solutions. In order to determine humus content, the total organic carbon (TOC) content was measured by standard wet chemistry technique. The measured soil sample was 0.5–1 g (MSZ 21 470-52:1983). The oxidation of organic materials was achieved by destruction with potassium dichromate (K_2_Cr_2_O_7_) in sulphuric acid (H_2_SO_4_), on the basis of the Walkley-Black procedure [44]. The chromium ions (Cr^3+^) were measured spectrophotometrically with an UV-1800 Shimadzu Spectrophotometer at 590 nm, because the amount of Cr^3+^ is in direct ratio with the amount of TOC. The humus content was calculated by the following equation: humus = TOC * 1.724. Carbonate content was determined with a Scheibler calcimeter based on a volumetric method [45] (ISO 10693:1995, IDT) (MSZ-08-0206-2:1978). Soil water capacity was determined using Arany-type plasticity index (PA), which was measured by deionized water titration [46] (MSZ-08 0205:1978). Total salt content was determined by measuring electrical conductivity with an Orion 5-Star conductivity meter from soil samples reaching the liquid limit during determination of PA (MSZ-08-0206-2:1978). The pH value (KCl and H_2_O) was measured electrometrically with Orion 5-Star conductivity meter [47] after 24 h (MSZ- 08- 0205:1978). Units of measurements were the following: PA: number without dimension; humus content: %; carbonate content and total salt dissolved in water: m/m %; others: mg/kg.

### 4.3. Nectar Sampling

Prior to nectar sampling, *A. ursinum* flower buds just prior to anthesis were covered with a mosquito net to exclude flower-visiting insects for 24 h before the measurement. Nectar was measured from 20–30 pollen-shedding flowers per individual, from 3 and 4 pots in 2013 and 2015, respectively, between 3:00 p.m. and 6:00 p.m. on each sampling day. Nectar was collected with the microcapillary method from the base of the ovary. We decided to apply this method, which is one of the standard nectar sampling procedures [13,14,18,48,49,50], because it proved to be reliable in our previous studies [27], and we wanted to compare our results with previous reports that also applied the microcapillary method [51,52]. The solute concentration (as % *w*/*w*, consisting almost totally of sugars) was measured using a handheld sugar refractometer ATAGO N-50E, Tokyo, Japan.

### 4.4. Statistical Analyses

Each nectar producing parameter of *A. ursinum* was analysed with linear mixed models using lm4-package [53]. For each model, year and site were treated as fixed factors and individuals were treated as random factor, to involve the individual differences among each examined plant. Log_e_ +1 transformation was made in the case of ratio of nectar-producing flowers. Transformation was based on graphical evaluation according to Crawley [54]. Models hypothesis testing was performed with Chi-square tests. For pair-wise comparisons, Tukey post-hoc tests were conducted in both cases with multcomp package [41] to compare the differences among all experimental set-ups.

To test how these measured soil features influence each nectar producing parameter of *A. ursinum*, multiple linear regressions were applied, using lm4-package [39]. In the models soil features were treated as fixed factors while sampling sites were treated as random factor to involve the differences among each examined population. Hypothesis testing was performed with Chi-square tests and for testing connection further, r^2^ values were calculated for the significant soil parameters.

## Figures and Tables

**Figure 1 plants-10-00597-f001:**
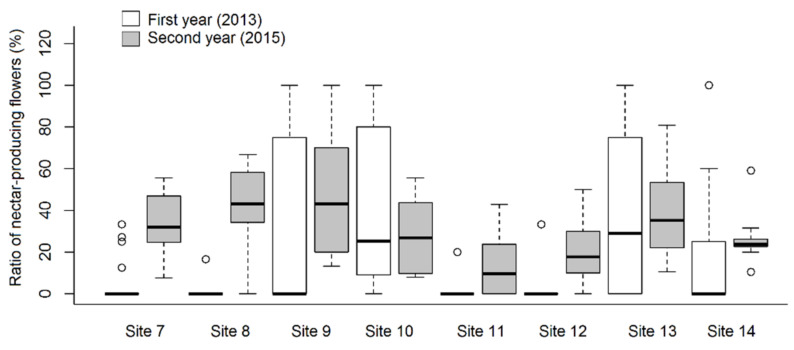
Ratio of nectar-producing flowers in *Allium ursinum* individuals at different study sites in two different years.

**Figure 2 plants-10-00597-f002:**
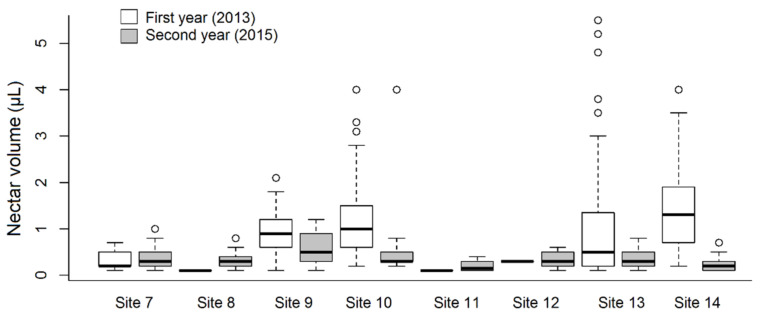
Nectar volume of individual *Allium ursinum* flowers at different study sites in two different years.

**Figure 3 plants-10-00597-f003:**
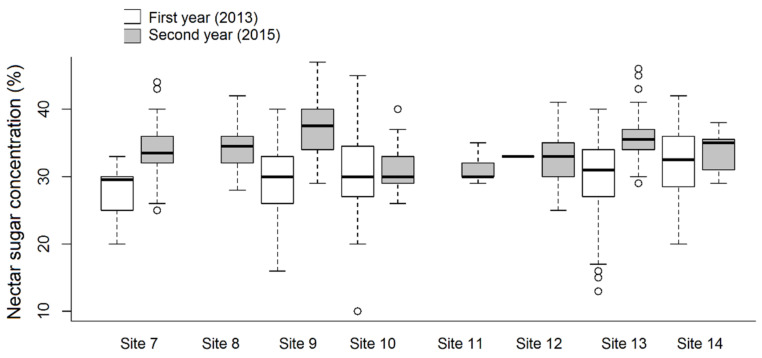
Nectar sugar concentration of individual *Allium ursinum* flowers at different study sites in two different years.

**Figure 4 plants-10-00597-f004:**
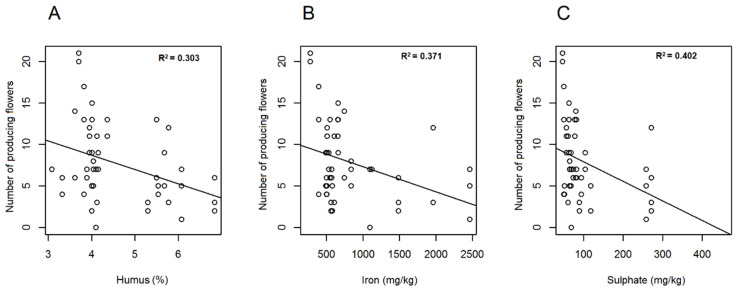
Linear regression of soil factors. Humus (**A**), iron (**B**) and sulphate (**C**) content of soil were plotted against the number of *Allium ursinum* flowers producing nectar.

**Figure 5 plants-10-00597-f005:**
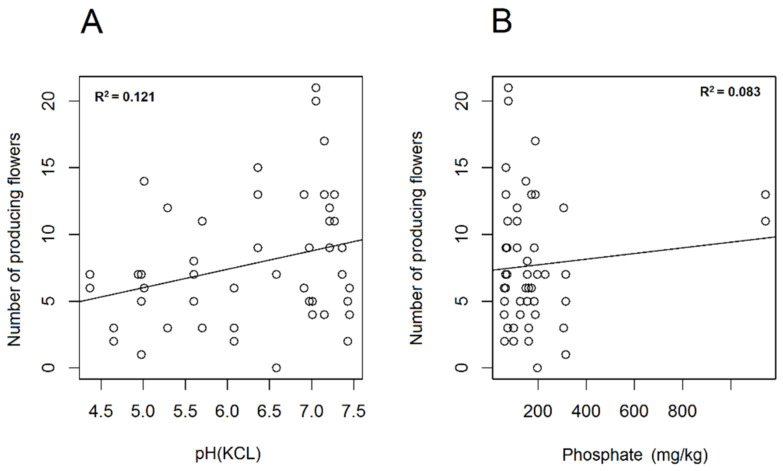
Linear regression of soil factors. pH (KCl) (**A**) and phosphate content (**B**) of soil were plotted against the number of *Allium ursinum* flowers producing nectar.

**Figure 6 plants-10-00597-f006:**
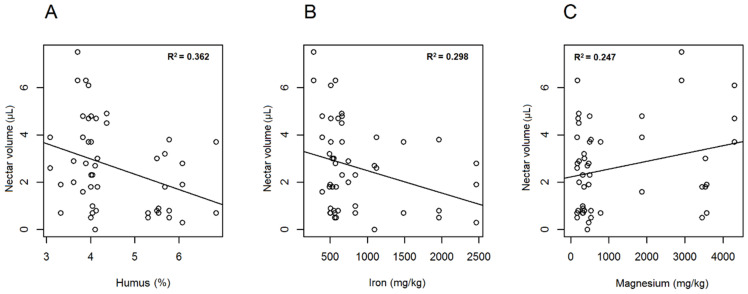
Linear regression of soil factors. Humus (**A**), iron (**B**) and magnesium (**C**) content of soil were plotted against the nectar volume in *Allium ursinum* flowers.

**Figure 7 plants-10-00597-f007:**
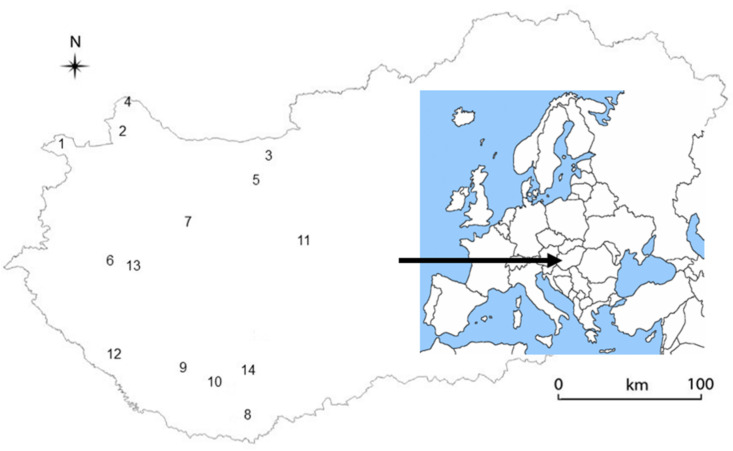
Map of Hungary showing study sites 1–14 of *Allium ursinum.*

**Table 1 plants-10-00597-t001:** Characteristics of sampling sites of *Allium ursinum*, with the total number of flowers sampled and the ratio of nectar producing flowers. Names of plant associations are listed according to Kevey, Borhidi et al. [28,29]. Soil types: BFS: brown forest soil; CFS: colluvium of forest soil; MC: mixed colluvium; HF: humic fluvisol; SF: soil of swamp forest. Soil analyses were done in 2013 at each site. In each sampling site, the coverage of *A. ursinum* was 80–100%/400 m².

Site #	Site Name	GPS Coordinates	Vegetation Type, Soil Type	Association	Number of Flowers Examined		Ratio of Nectar Producing Flowers (%)	
					2013	2015	2013	2015
1	Ágfalva	n47°41.253 e16°31.372	oak-hornbeam forest, BFS	*Cyclamini purpurascenti-Carpinetum*	207	-	19.3	-
2	Jánossomorja	n47°46.518 e17°9.189	hardwood gallery forest, HF	*Pimpinello majoris-Ulmetum*	64	-	82.8	-
3	Pusztamarót	n47°41.032 e18°31.872	beech forest, BFS	*Daphno laureolae-Fagetum*	143	-	6.3	-
4	Rajka	n48°496 e17°12.622	hardwood gallery forest, HF	*Pimpinello majoris-Ulmetum*	235	-	17.4	-
5	Tatabánya	n47°31.971 e18°25.835	oak-hornbeam forest, BFS	*Corydalido pumilae-Carpinetum*	148	-	0.0	-
6	Zalaistvánd	n46°92.813 e17°139	oak-hornbeam forest substitute for black locust, BFS	*Corydalido pumilae-Carpinetum Robinia pseudoacacia consoc.*	147	-	7.5	-
7	Bakonybél	n47°30.638 e17°69.428	beech forest, BFS	*Daphno laureolae-Fagetum*	132	114	5.3	38.6
8	Bisse	n45°89.981 e18°27.6851	oak-hornbeam forest, BFS	*Asperulo taurinae-Carpinetum*	126	97	0.8	55.7
9	Bőszénfa	n46°13.781 e17°51.984	oak-hornbeam forest, BFS	*Helleboro dumetorum-Carpinetum*	172	119	29.7	25.2
10	Lapis	n46°7.304 e18°12.073	ravine forest, CFS	*Scutellario altissimae-Aceretum*	247	199	39.7	26.6
11	Lórév	n47°6.545 e18°53.566	hardwood gallery forest, HF	*Scillo vindobonensis-Ulmetum*	174	152	0.6	3.9
12	Szenta	n46°22.739 e17°24.306	alder gallery forest, SF	*Aegopodio-Alnetum glutinosae*	127	126	1.6	23.0
13	Zalaszántó	n46°87.145 e17°21.684	alder gallery forest substitute for black walnut, SF	*Aegopodio-Alnetum glutinosae Juglans nigra consoc.*	213	170	39.0	35.9
14	Zobákpuszta	n46°11.658 e18°19.066	transition of oak-hornbeam wood and alder gallery forest, MC	*Asperulo taurinae-Carpinetum et Carici pendulae-Alnetum glutinosae confer*	212	62	20.8	19.4

## Data Availability

Voucher specimens of *Allium ursinum* L. from each study site were deposited in the Herbarium of Department of Plant Biology, University of Pécs. The data presented in this study are available within the article and in Appendix A.

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
