# Peer review of "Soil Humus, Iron, Sulphate and Magnesium Content Affect Nectar Traits of Wild Garlic (Allium ursinum L.)"

_plants, 2021, doi:10.3390/plants10030597_

Round 1
Reviewer 1 Report
Please see attachment

Reviewer 2 Report
Ref. Soil humus, iron, sulphate and magnesium content affect nectar traits of wild garlic (Allium ursinum L.)
In this manuscript entitled ‘Soil humus, iron, sulphate and magnesium content affect nectar traits of wild garlic (Allium ursinum L.)’ the authors present significant differences in several nectar traits depending on a vulnerable soil properties. Studies such as this one that measure nectar volume and concentration on the background of soil properties are fairly rare. The results have ramifications for beekeepers who decide on their colony movements.
While I do have some criticisms of some methods for nectar sampling and the perspective in which some of the data are presented, I found this ms to be well-written and clear.
In particular, I have some uncertainty concerning the soil properties (no data presented).
I see not much revisions to improve this ms and make it a stronger article. I offer the following comments to assist the authors in improving this ms.
My major concern
The table showing the difference in soil properties between study site is required. Such data should be added as supplementary material.
Moreover, if the plants for nectar sampling were kept under controlled environmental conditions, please add more specific information on these conditions (e.g. temperature, air humidity)
Introduction
Please, add to the references more up-to-date literature on nectar volume, concentration and sugar composition, e.g.
Bożek M., 2019. Nectar secretion and pollen production in Hyacinthus orientalis ‘Sky Jacket’ (Asparagaceae). Acta Agrobotanica. 72(4): 1796. https://doi.org/10.5586/aa.1796
Denisow B. , Strzalkowska-Abramek M. , Wrzesien M. 2018.
Nectar secretion and pollen production in protandrous flowers of Campanula patula L.(Campanulaceae) Acta Agrobotanica. 71(1), 1734.
Antoń S., Komoń-Janczara E., Denisow B. 2017. Floral nectary, nectar production dynamics and chemical composition in five nocturnal Oenothera species (Onagraceae) in relation to floral visitors Planta. 246(6):1051-1067. doi: 10.1007/s00425-017-2748-y.
Results
Table 1. Last column – is Producing flower rate – it is not clear; please make correction, e.g. The rate of nectar producing flowers
Figure 1 – this same comment ; it is not clear that you mean of nectar producing flowers. The rule is that all descriptions should be self- descriptive
Results Figs. 4-6. Please, add what correlation type have been made.
- 236 - Fig 7. If the map would be given on the background on the map of Europe, it would be more readable and unmistakable for not European readers of your ms.
Discussion
L218 – Wild garlic habitats ??? – it is not precise. What does it mean
Material and Methods
L 294-302- the nectar sampling is not clear for me.
Why do you use the covering the plants with nets, if the plants were kept in a growth room. Am I right, that the sampling plants for nectar production were not kept outside? What floral visitors you expected in laboratory.?
Moreover, how flowers were marked? How do you recognized 24-old flowers in umbel-like inflorescence of A. ursinum?
The nectar sampling methods have to be described in more details.
The table showing the difference in soil properties between study site is required. Such data should be added as supplementary material.
Moreover, if the plants for nectar sampling were kept under controlled environmental conditions, please add more specific information on these conditions (e.g. temperature, air humidity)
L. 228 – if A. ursinum is dominant, please add this information to the Table 1, giving the % of the cover e.g. in releves
Reviewer 3 Report
The manuscript titled Soil humus, iron, sulphate and magnesium content affect nectar traits of wild garlic (Allium ursinum L.) is a useful contribution for plants.
Some concerns that authors should consider.
- The introduction section is very precise. It needs elaboration around a clear hypothesis.
- Results are very redundant and should be carefully written with only including the important and useful information.
- Figure 4,5, and 6 are of weak resolution.
- Elaborate the discussion section with a correlation of the studies with your work.
- I wonder what the effect of these substances on cultivated allium species? Generally, wild species have better nector for survival in the wild.
- Add a conclusion section as well.
- Carefully check language mistakes.
Round 2
Reviewer 3 Report
Manuscript can be accepted.